# Preserving Sunken Military Vessels as Underwater Cultural Heritage in Colombia: Legal Challenges and Prospects for the USS *Kearsarge* Wreck Site

**William Gomez Pretel** [1] , **Moonsoo Jeong** [2,*] , **Camilo Ernesto Rodríguez-Gutiérrez** [3] **and Agustin Ortiz JR** [4]

1   Independent Researcher, Taejong-ro 702, Eden Kumho, 101-1103, Yeongdo-gu, Busan 49122, Republic of Korea; wgomezp77@gmail.com
2   Division of Navigation Convergence, Korea Maritime and Ocean University, Busan 49112, Republic of Korea
3   Veeduría Ciudadana por la Equidad Fiscal, Calle 12 # 5-32, Bogota 111631, Colombia; camiloe.rodriguez@urosario.edu.co
4   The Federal Emergency Management Agency, 500 C ST SW, Washington, DC 20472, USA; 19agustin19@gmail.com
*   Correspondence: jms@kmou.ac.kr

**Abstract:** This study examines the legal challenges related to preserving sunken military vessels as Underwater Cultural Heritage (UCH) in Colombia. These challenges include Spanish galleon shipwrecks, limited international cooperation, and the lack of legal recognition for sunken military vessels under domestic law (Law 1675 of 2013). To address these issues, this article reviews the concepts of warship and sovereign immunity as they relate to the status of sunken military vessels. The study places a particular focus on the USS *Kearsarge*, a military shipwreck in Colombian territorial waters protected by the Sunken Military Craft Act of 2004 (SMCA) of the United States. Additionally, it analyzes the legal frameworks and management of UCH in both Colombia and the United States, as well as providing two lists of Colombia's sunken military vessels and foreign sunken military vessels in its waters. The research concludes by highlighting the complexities of managing UCH in Colombia and offering a prospectus on the future of the USS *Kearsarge* wreck site as shared heritage. Ultimately, this study underscores the need for a more comprehensive legal framework and greater international cooperation to ensure the preservation and protection of sunken military vessels in Colombia.

**Keywords:** sunken military vessels; underwater cultural heritage; Colombia; United States; USS Kearsarge; management and preservation

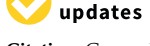



## 1. Introduction

Colombia's rich history of shipwrecks, including numerous military vessels [1], presents significant challenges for the management and preservation of Underwater Cultural Heritage (UCH) due to uncertainties surrounding their status and sovereign immunity [2]. These challenges are exacerbated by Spanish galleon shipwrecks, claims of past colonial exploitation, limited international cooperation, and the absence of recognition of sunken military vessels under the Submerged Cultural Heritage Law 1675 of 2013 [3–5]. The case of the Spanish galleon *San Jose*, which sank near Cartagena de Indias in 1708 following a battle with the British fleet [6,7], is particularly contentious. The official discovery of the *San Jose* by the Colombian government in 2015 [8] has sparked controversy regarding its possible commercial salvage by a public–private association, which adds to the legal dispute between the Colombian government and an American salvage company that claims to have found the remains in 1982 [9]. Furthermore, Colombia solely regulates the management of its underwater heritage through domestic law, as it is not a party to the United Nations Educational, Scientific and Cultural Organization (UNESCO) 2001 Convention on

the Protection of the Underwater Cultural Heritage or the United Nations Convention on the Law of the Sea (UNCLOS) [10]. Although several recent studies have focused primarily on the legal case of the *San Jose* as a sunken military vessel and criticized the Submerged Cultural Heritage Law of 2013 [2,3,9,11–15], there is a literary gap in the management of sunken military vessels from other countries in Colombia.

The purpose of this article is to explore the legal and managerial implications of sunken military vessels in Colombia's territorial waters. Specifically, it will examine the case of the USS *Kearsarge*, which sank in Roncador Cay, Colombia, in 1894 while en route to protect American interests in Central America. As the flagship of the North Atlantic Command, the vessel was a significant naval asset of the United States [16]. Today, the United States recognizes the sovereign immunity status of the USS *Kearsarge*'s remains under the Military Craft Act of 2004 (SMCA), while Colombia categorizes the vessel as an underwater cultural heritage asset under Law 1675 of 2013. To achieve its objectives, this study will review the concept of warship and sovereign immunity under both international and domestic laws in Colombia and the United States. Additionally, it will describe the legal framework and current state of underwater cultural heritage from sunken military vessels in both countries over the past years. This research will also explore the management and status of UCH and recent practices concerning the preservation and management of sunken military vessels in light of their respective national interests. In addition, it will identify the maritime interests of both countries and their relationship with sunken military vessels [17]. The article will provide a list of Colombian sunken military vessels and foreign sunken military vessels in Colombian waters, with the goal of facilitating a better understanding of maritime interests. The database highlights the UCH from sunken military vessels shared between Colombia and other countries and helps to interpret the magnitude of its national interests and naval power size. Sunken Spanish military vessels are, however, excluded due to the ambiguity surrounding their involvement in commercial activities. Nonetheless, Act 14 of 2014 considers all sunken state vessels from Spain as publicly state-owned assets that enjoy sovereign immunity, further complicating their status in Colombian waters [4,18]. The study emphasizes the necessity of cooperation between flag states and coastal states to safeguard and preserve sunken military vessels in Colombia. Furthermore, the research highlights the challenges and complexities of managing UCH in Colombia and offers a prospectus on the future of the USS *Kearsarge* as shared UCH. The article underscores the need for a more comprehensive legal framework and greater international cooperation in Colombia to ensure the preservation and protection of sunken military vessels as cultural heritage assets. By offering a novel perspective on the legal and management issues surrounding sunken military vessels in Colombia, this article provides a valuable contribution to the scholarly literature on maritime law and underwater cultural heritage management.

## 2. Legal Status and Immunity of Sunken Military Vessels

The presence of sunken military vessels within territorial seas raises a complex legal issue that revolves around uncertainties regarding their status as warships and their engagement in military operations during their sinking. Moreover, the differing interpretations of international law by flag and coastal states add further complexity to this matter [19]. To gain a thorough comprehension of the sovereignty rights of flag and coastal states, it is crucial to examine the legal entitlements of state vessels, particularly warships, within the framework of international law, encompassing aspects such as ocean division and boundary status.

The United Nations Convention on the Law of the Sea (UNCLOS) of 1982 provides a framework for determining the legal status of the territorial sea and establishing sovereignty rights. The legal provisions for this are explicitly outlined in Part II–Territorial Sea and Contiguous Zone, specifically in Articles 2 and 3. These provisions limit the territorial sea to 12 nautical miles, conferring sovereignty and territorial jurisdiction upon the coastal state within this area. Additionally, UNCLOS outlines other legal boundaries in Part V–Exclusive

Economic Zone, Part VI–Continental Shelf, and Part VII–High Seas [10]. The concept of a warship, as defined by customary international law, has been further elaborated in UNCLOS. Article 29 of the Convention specifies that a warship is a ship that belongs to the armed forces of a State, bears the external marks that distinguish such ships of its nationality, is under the command of an officer commissioned by the government of the State, and is manned by a crew that is under regular armed forces discipline [10]. In addition to providing a definition of warships, UNCLOS also clarifies the concept of sovereign immunity for state-operated ships. Articles 32, 96, and 236 of the Convention grant sovereign immunity to warships and other types of state vessels, as long as they are not engaged in commercial activities [10]. It is essential to note that the UNCLOS provisions on warships and sovereign immunity have become customary international law and are binding on all states, regardless of whether they are parties to the Convention, as in the case of Colombia and the United States [20,21]. However, the legal status and immunity of sunken military vessels are not explicitly referenced in UNCLOS, and their immunity when they are sunk is subject to debate. While sunken warships are still considered property, they no longer fall under the command of a duly commissioned officer and are unable to navigate. Consequently, some argue that they may not be classified as "warships" and are thus no longer subject to the international regime [19,22]. As a result, the legal status of sunken military vessels remains uncertain, leading to ongoing discussions and disagreements. Maritime powers claim that sunken military vessels and their artifacts continue to enjoy sovereign immunity, even when located in the territorial waters of coastal states.

This perspective, however, remains a contentious and highly debatable issue [23–25]. Ownership disputes over sunken military vessels within the jurisdiction of coastal states have arisen, particularly when former colonial powers or blue-water navies assert rights over them [26]. The legal status of such vessels remains contentious, as some argue that they only lose their legal status through recognized practices such as surrender or capture during battle before sinking, gift, or abandonment of the property. There is, however, no legal or international consensus on the role of the flag state or criteria for granting legal status to a wreck [19,25,27]. Furthermore, the extraction of resources during the colonial era has become increasingly controversial, as certain local communities assert their rights to these cultural resources [28]. Over time, the exploitation of the colonial past has also led to challenges from former colonial powers, who make claims under the criteria of international law and sovereign immunity rights [13,26,29]. Additionally, the conflict between preserving the archaeological context and the perception of "valuable cargo" by commercial salvage companies and other stakeholders on certain sunken military vessels further complicates the discussion. The absence of established criteria for defining archaeological or historical elements of interest means that the interpretation of the law is contingent upon the interests of the involved states [26].

Considering the ongoing disputes surrounding sunken military vessels, technological advancements in underwater exploration have facilitated salvage or recovery operations. Some salvage operations have demonstrated adherence to international and state practices, as well as appropriate methods and respect for underwater cultural property, war graves, and military shipwrecks through bilateral agreements and cooperation [26,30,31]. Nevertheless, apprehensions have been raised regarding certain salvage operations that prioritize economic gain over cultural heritage preservation, disregarding archaeological standards and engendering diplomatic incidents [24,32,33]. Consequently, navigating the intricate legal frameworks governing sovereign immunity and salvage operations of sunken military vessels under international law remains uncertain. To address this gap, the 2001 UNESCO Convention was created. However, maritime powers such as the United States, the United Kingdom, and other nations have also established their own domestic laws pertaining to the protection of sunken warships. Additionally, bilateral agreements, international cooperation, and case law all play crucial roles in addressing concerns related to the protection and management of UCH [27,34,35].

### 3. Sunken Military Vessels and Underwater Cultural Heritage

The 2001 UNESCO Convention stands as the most significant international legal instrument dedicated to the protection and sustainability of underwater cultural heritage (UCH) [36]. However, protecting sunken military vessels poses a complex and ambiguous challenge within the framework of this Convention. While the primary objective of the Convention is to safeguard sites of historical and archaeological importance, including shipwrecks [19], determining the eligibility of sunken vessels for protection under international law remains a critical issue. According to Article 1(1)(a) of the 2001 UNESCO Convention, all submerged traces of human existence with cultural, historical, or archaeological significance that have remained underwater, either partially or completely, for a minimum of 100 years are eligible for protection, encompassing vessels and their associated artifacts [36]. However, applying these criteria to sunken warships, particularly those that sank in the territorial waters of foreign coastal states and have not yet reached the 100-year threshold, raises questions about their status as UCH [19,22]. In addition to the lack of consensus within the international community regarding the definition of UCH, the UNESCO Convention fails to address the principles of sovereign immunity concerning sunken warships or state-owned vessels, further complicating the matter [22]. Despite being the most prominent international legal instrument in this domain, the UNESCO Convention has not gained widespread recognition, with only 72 countries having ratified it to date [37]. Additionally, certain countries such as Colombia, despite being signatories, are yet to ratify it, while the United States is not a signatory at all [9,35]. The limited acceptance and implementation of the Convention globally add to the challenges in protecting sunken warships, especially within the territorial waters of other nations.

Moreover, the complexities surrounding UCH are compounded by vested interests, particularly with regard to recognizing a third state's rights to sunken warships in its territorial waters, their identification, and their eligibility for protection under national law and the UCH status [26]. The lack of a comprehensive legal framework governing UCH adds to the challenge, compounded by the existence of multiple legal standards, whether multilateral, bilateral, or unilateral, resulting in further uncertainty. This complexity is particularly evident in the case of sunken military vessels, where no specific regulations exist to simplify the legal structure [24]. Additionally, jurisdiction and ownership present a dilemma in protecting underwater cultural heritage, as sovereign immunity creates ambiguity over territorial control, while flag states claim ownership of the vessel. In addition, former colonial powers or blue-water navies may assert claims over war graves to prevent interference [19]. Despite these complexities, war graves are protected under the Convention and should be treated with utmost respect [38].

There are challenges associated with determining the correct legal and relevant norms in this situation, but complexities and an absence of normativity are making this problematic. Furthermore, customary law on sunken military vessels further adds to the challenge of determining the correct legal framework for UCH. The complexity of interpreting national laws on UCH in the context of sunken military vessels is compounded by the influence of national interests and the size of a country's maritime power.

### 4. Sunken Military Vessels in Colombia and the United States

To gain a better understanding of the challenges associated with protecting sunken military vessels as part of underwater cultural heritage, this research aims to analyze several key aspects. Specifically, this includes an examination of the relevant domestic laws, the naval heritage of sunken military vessels, and their current state in both Colombia and the United States. This analysis presents a valuable opportunity to further examine the case of the USS *Kearsarge*, a United States military vessel that was wrecked in Colombian territorial waters in 1894.

*4.1. The United States Sunken Military Vessels*

The origins of the United States as a maritime state with a focus on sea power have resulted in an extensive inventory of sunken military vessels. This strategic approach helped establish one of the largest navies in the world, a legacy that persists today [39]. The US government carefully protects sunken military vessels and wreck sites, spanning both peace and war times, resulting in one of the largest collections globally, surpassed only by the United Kingdom [40,41]. This potential, viewed from a heritage perspective, has led to numerous discoveries relating to different periods in American naval history and has generated extensive inventories and databases [42–54]. This extensive collection of studies highlights the fundamental role of marine culture, providing insights from various disciplines.

Sunken military vessels have been present in the US since the American Revolutionary War, which saw the establishment of the "Continental Navy." However, it was not until the Naval Act of 1794 that the US Navy was officially founded by John Adams with the first ships [55,56]. Sunken military vessels span both the US maritime domain and beyond, including those from colonial times, particularly from the American Revolutionary War. Furthermore, military shipwrecks from the nineteenth century can be identified, ranging from the War of 1812 to the American Civil War, with some wrecks located outside of US maritime territory [45]. This period was significant for sunken military vessels as the United States developed a blue-water navy and a new doctrine, aimed at influencing global diplomacy [55,57]. Additionally, there are numerous sunken military vessels from the twentieth century, including those from World War I and, more notably, World War II. These wrecks encompass a significant component of sunken military ships, state vessels, and gravesites at sea in the United States [58] and worldwide [41].

The United States is home to approximately 3000 shipwrecks from both domestic and foreign vessels, including military shipwrecks, lying within its national marine sanctuary, as identified by the National Oceanic and Atmospheric Administration (NOAA) [59]. Over 400 of these sites have been protected by NOAA, exemplifying the nation's commitment to preserving its underwater military heritage. The USS *Monitor*, for instance, became the first national marine sanctuary in 1975 [60]. To this end, NOAA has established and protected 15 marine areas, covering more than 620,000 square miles of ocean and Great Lakes waters [45,61]. Although NOAA has military wreck sites within their sanctuaries, the US Navy's Naval History and Heritage Command (NHHC) assumes principal responsibility as the custodian of these resources for the US Navy. The NHHC has been extensively involved in conducting research and implementing preservation initiatives for approximately 2500 US sunken military vessels worldwide [61–63]. The US government's commitment to preserving its military heritage is evident in recent technological advancements that have led to the discovery of several sunken wrecks [64]. Notably, the USS *Samuel B. Roberts*, found at a depth of 6895 m, marked the deepest wreck ever located [65], while the USS *Albacore* was located in collaboration with Japan in 2023 [66]. Numerous military wrecks have also been identified, such as the USS *Somers*, USS *Cumberland*, USS *Tecumseh*, CSS *Alabama*, USS *Huron*, USS *Arizona*, USS *San Diego*, USS *Conestoga*, USS *Houston*, USS *Macon*, and USS *Indianapolis*, among others [64,67]. The Naval History and Heritage Command has also conducted archaeological surveys of specific wreck sites resulting from Operation Neptune during World War II, including Omaha Beach, Utah Beach, Banc du Cardone, Pointe du Hoc, and Pointe et Raz de la Percée [68].

These recent discoveries of military wrecks have brought attention to the vast number of sunken vessels that still remain in the oceans, particularly from World War II [69]. Furthermore, the preservation and protection of war graves, such as those of the USS *Lagarto*, USS *Wahoo*, and USS *Grunion*, is a critical concern in the United States [35,68,70]. In addition, the archives and records related to military wrecks are essential for historical analysis and interpretation of archaeological findings [71,72]. The United States government maintains primary source records related to shipwrecks in the National Archives and Records Administration (NARA) as well as in the archives at Naval History and Heritage

Command, which compile information from thousands of records, including logbooks, courts-martial, and naval officers' reports for the Navy and US Coast Guard Records [73,74]. Similarly, the Library of Congress (LOC) also houses a collection of classified records pertaining to state vessels and foreign military wrecks in US territory [75]. While the NHHC's comprehensive wreck database remains inaccessible to the general public due to sensitivity with location information to prevent looting of war graves and/or commercial exploitation, the NHHC also maintains primary sources which are publicly accessible through its Archive, Library, Underwater Archaeology Branch, and Museum network at the Washington Navy Yard in Washington DC. Other resources such as the US Navy Losses in World War I, showcasing interactive maps, photographs, and estimated locations of losses, are also available [76].

The United States' commitment to maritime heritage and nautical archaeology is evidenced by its status as one of the world's foremost maritime powers, as well as the home of the world's largest navy [77].

### 4.2. Colombia Sunken Military Vessels

Colombia possesses a vast potential for underwater cultural heritage from shipwrecks, with a significant portion located in its territorial waters [12,78]. The Spanish colonial period in particular, marked by the extensive naval presence of Spain and Britain, has left behind notable sunken military vessels. Many of these wrecks are believed to be located in the waters between Cartagena de Indias and Havana, with the Archipelago of San Andres, Old Providence, and Santa Catalina suspected to have been a ship trap [79–81]. Spain has around 2000 naval wrecks worldwide, many of which are located in the Colombian Caribbean [82–87]. Although the Spanish government is currently undertaking a project to classify and inventory its sunken naval vessels worldwide, including those in Colombian waters, the database for these wrecks is not yet accessible to the public [88,89]. The presence of sunken military vessels in Colombia is not limited to Spanish wrecks, as the United Kingdom, the United States, and the Netherlands also have a few military shipwrecks in Colombian waters (see Table 1). A noteworthy database in this regard is the Royal Navy Lost List [90], which provides comprehensive information on naval wrecks from 1512 to 2004, including those situated in Colombia, among other sources [43,46,91–95].

Notably, Colombia's inventory or database of sunken military vessels, regardless of their origin, remains inaccessible to the general public. To address this gap in knowledge, this study classified foreign sunken warships in Colombian waters and compiled them into a list to serve as a resource for further research (Table 1). Nonetheless, the classification of military or commercial operations can be challenging when examining the Spanish naval fleet activities during the colonial period. In particular, the categorization of specific Spanish sunken military vessels remains unclear, necessitating careful evaluation of the circumstances surrounding their sinking [81,96,97]. For example, the galleons *San Jose* and *San Roque* pose a significant challenge to the classification process, as their missions appear to have involved both military and commercial activities. The *San Roque*, which wrecked along with part of the Spanish Armada fleet near Serrana Bank and Roncador Cay in 1605, was reportedly engaged in both military and commercial activities [98]. Similarly, the *San Jose* was transporting valuable commercial cargo when it sank in 1708 during a naval encounter with a British fleet near Cartagena de Indias, raising questions about its main mission [6,7]. These uncertainties underscore the challenges associated with identifying whether a naval vessel can be classified as strictly military or commercial. Consequently, to ensure clarity and consistency in the research, only foreign vessels with a clear military mission were included in our exhaustive list of sunken military vessels in Colombia, while Spanish vessels were excluded.

**Table 1.** List of foreign military vessels that have been recorded as sunk in Colombian waters.

| No | Name | Year of Loss | Nation | Location | Sources |
|----|------|--------------|--------|----------|---------|
| 1 | Unidentified | 1675 | Netherlands | Roncador Cay | [79] |
| 2 | HMS *Jersey* | 1707 | United Kingdom | Punta Canoa | [90,94] |
| 3 | *Galicia* | 1741 | (Spanish) captured by United Kingdom | Cartagena Harbor | [99–103] |
| 4 | HMS *Legere* | 1801 | (French) captured by United Kingdom | Galerazamba | [90,94] |
| 5 | HMS *Raposa* | 1808 | (Spanish) captured by United Kingdom | Rosario Islands | [94,95] |
| 6 | HMS *Jackdaw* | 1835 | United Kingdom | Old Providence Island | [104,105] |
| 7 | USS *Kearsarge* | 1894 | United States | Roncador Cay | [16] |
| 8 | USS *Peacock* | 1940 | United States | Cartagena Harbor | [106] |

Note: Sunken military vessels from Spain have been excluded from this inventory.

Although Colombia is a country facing both the Pacific Ocean and the Caribbean Sea, it has not traditionally been considered a major maritime nation, resulting in limited naval wrecks because of the size of its navy and national interests [39,107]. The Colombian Navy, officially known as the *Armada República de Colombia* (ARC) in Spanish, has a modest naval heritage compared to other maritime powers such as the United States or large blue-water navies. The ARC has military sunken vessels from the early nineteenth to twentieth-first centuries, encompassing significant historical events such as the Independence War and the War of a Thousand Days (1899–1903) [108]. During the War of Independence, the main naval force in Cartagena de Indias was commanded by Admiral Jose Padilla, one of the few Colombia naval heroes who fought in the Battle of Trafalgar [109]. However, there are no official records of shipwrecks during the War of Independence from the "Patriotic Navy of Gran Colombia" [110] due to the loss or disappearance of files during the conflict [111]. Despite the fact that some records exist, the fate of ships involved in the independence conflict remains ambiguous. Although figures such as Luis Brion and Louis-Michel Aury were appointed captains (*capitán de navío* in Spanish) and received official support and military ranks, the lack of recognition of privateers as official warships during the war of independence has contributed to this uncertainty [112–114]. The ambiguity is exemplified by the wrecks of privateers such as Louis-Michel Aury and some of its ships, which were lost during a hurricane in 1818 on Old Providence Island [113,115]. The classification of these privateer wrecks as sunken military vessels remains uncertain, as they did not have the same legal status as official warships [116]. Additionally, during the War of Independence in Colombia, some American privateers provided support, particularly in Cartagena de Indias, which may have resulted in potential wrecks of American vessels in Colombian waters [117,118]. Although Colombia gained its independence in the early nineteenth century after the naval Battle of Maracaibo [119], it was not until the subsequent century that a formal Navy, including a Naval Academy, was established in response to the 1932 conflict with Peru [120,121]. One essential element of Colombia's naval heritage is its significant contribution to the Korean War as part of the United Nations Task Force 95 from 1951 to 1955. However, it is worth noting that Colombia did not register any shipwrecks during this conflict [122,123].

One of the objectives of this study is to classify the sunken military vessels of Colombia, resulting from various causes such as naval warfare, military exercises, grounding, intentional sinking for the establishment of natural parks, and those located outside Colombia's jurisdiction. This classification will provide valuable insights into Colombia's maritime power and enable meaningful comparisons with the naval wrecks of other countries (Table 2). Despite the importance of naval wrecks as cultural heritage under Colombian law, only a limited number of sunken military vessel wreck sites have been investigated in recent years. For example, the 1741 British attack on Cartagena de Indias resulted in the

sinking of numerous Spanish warships, including the *Conquistador*, *San Carlos*, *San Felipe*, *Dragon*, and *Africa* [124–126]. The potential identification of these sunken vessels has been the subject of extensive research, such as the work undertaken by Del Cairo et al. [127,128] and Quintana-Saavedra et al. [129,130]. Additionally, the Colombian Navy has located and identified the USS *Peacock* (AM-46), a US Navy ship that sank in Cartagena's harbor after a marine accident [131,132]. In 2015, the Colombian government officially discovered the Spanish galleon *San Jose*, but it has not yet received significant research attention [8].

The scarcity of research on sunken military vessels in Colombia reflects the country's slow development in maritime science and technology. This can be attributed, in part, to a lack of international cooperation. However, recent developments, such as the establishment of the Ministry of Science and Technology [133] and the new subsea capabilities of the Colombian Navy [8], are expected to drive significant progress in research efforts in this field. Furthermore, a series of remarkable projects has emerged, highlighting the vital importance of institutional cooperation among state agencies, local communities, and stakeholders [134–136]. These recent initiatives effectively demonstrate the tangible benefits derived from cooperative efforts across different stakeholder groups and serve as compelling examples for future research on military sunken vessels in Colombia.

**Table 2.** List of Colombian military vessels that have been recorded as sunk.

| No | Name | Year of Loss | Location | Sources |
|---|---|---|---|---|
| 1 | *Marte* | 1823 | Lake Maracaibo, Venezuela | [137] |
| 2 | *Gran Bolivar* | 1823 | Lake Maracaibo, Venezuela | [138] |
| 3 | *Rayo* | 1867 | Cartagena Harbor, Colombia | [139,140] |
| 4 | *Cuaspad* | 1867 | Trinidad and Tobago | [141,142] |
| 5 | *General Maza* | 1877 | Nassau Harbor, Bahamas | [143] |
| 6 | *La Popa* | 1901 | Between Galerazamba and Savanilla, Colombia | [144,145] |
| 7 | *Lautaro* | 1902 | Naos Island, Panama | [146,147] |
| 8 | *Boyaca* | 1903 | Panama | [148,149] |
| 9 | *Bogota* | 1936 | Cartagena Harbor, Colombia | [150] |
| 10 | *Cordoba* | 1937 | Deliberately sunk in Salmedina Bank, Colombia | [150] |
| 11 | *General Mosquera* | 1944 | Cartagena Harbor, Colombia | [150] |
| 12 | *Boyaca* | 1944 | Near Cuba | [150] |
| 13 | *Almirante Padilla* | 1964 | Bolivar Cay, Colombia | [150] |
| 14 | *Almirante Brion* | 1972 | La Guajira, Colombia | [150] |
| 15 | *Bahia Honda* | 1975 | San Andres Island, Colombia | [150] |
| 16 | *Sebastian de Belalcazar* | 2004 | Intentionally sunk in the Colombian Pacific Ocean | [150] |
| 17 | *Pedro de Heredia* | 2007 | Intentionally sunk in front of Tierrabomba Island, Colombia | [150,151] |
| 18 | *Quindío* | 2015 | Intentionally sunk in Ciénaga de los Vásquez, Baru Island, Colombia | [152] |
| 19 | *Pascual de Andagoya* | 2017 | Colombian Pacific Ocean | [153] |
| 20 | *Cartagena de Indias* | 2019 | Deliberately sunk in the Archipelago of San Bernardo, Colombia | [154,155] |
| 21 | *Buenaventura* | 2019 | Deliberately sunk in the Archipelago of San Bernardo, Colombia | [154,155] |

*4.3. World War II Sunken Ships in Colombian Waters*

During World War II, a few Colombian commercial vessels were lost due to attacks by German submarines. The Colombian schooner *Resolute* was sunk near Old Providence Island in 1942, presumably by the German submarine *U-159*. Additionally, the schooner *Roamar* was sunk while sailing between San Andres and Cartagena, likely by the *U-505*,

while the schooner *Ruby* was sunk in 1943 near San Andres Island, apparently by the *U-516* [120,156–160]. Additionally, the US Merchant Marine Corps played a vital role in providing logistical support to military operations during World War II. These vessels were, however, vulnerable to attacks, with German submarines targeting and sinking them in the Caribbean, including Colombian waters [161]. Among the losses was the steam tanker *Esso Harrisburg*, which was torpedoed and sunk, possibly by the German submarine *U-516*, in July 1944, approximately 82 miles north of Cabo de la Vela, Colombia. The attack resulted in the loss of four out of 28 Armed Guard crew members from the US Navy [162,163]. Similarly, in February 1942, the American steam tanker *J.N. Pew* was torpedoed by the German submarine *U-502*, about 96 miles north of Santa Marta, Colombia, subsequently sinking [161,162]. In September of the same year, the Canadian Steamer *John A. Holloway* was also attacked and sunk possibly by the German submarine *U-164* while sailing in convoy GAT.2, approximately 190 miles north of Punta Gallinas, Colombia [159].

Although the Armed Guards and other armed merchantmen played a crucial role during the war, with an impressive legacy, they did not enjoy sovereign immunity for engaging in commercial activities. Nevertheless, the loss of sunken vessels during the War in Colombian waters remains a significant aspect of Colombia's nautical history and underwater cultural heritage.

## 5. Domestic Laws in Colombia and the United States

### 5.1. Colombia's Domestic Laws

Despite Colombia's geographical advantage of having coasts in both the Caribbean Sea and Pacific Ocean, with islands and an archipelago in the Caribbean [164], its maritime power and regulatory framework differ significantly from that of countries such as Spain or the United States. The absence of a clear procedure for the protection of underwater cultural heritage and management of military shipwrecks can be attributed to a slow maritime evolution caused by a centralist and Andean strategic policy. This policy did not prioritize the development and utilization of the country's maritime potential, despite its strategic location near the Isthmus of Panama [13,107,165,166]. Moreover, the location of Colombia's capital city, Bogota, 2600 m above sea level in the Andes Mountains, far from its principal port cities, poses a significant challenge for the development and implementation of effective maritime policies [123,167,168]. However, the Colombian government has recently undergone a strategic reorganization to develop a robust maritime strategy, with the aim of positioning the country as a regional leader in line with the concept of Sea Power. These efforts have led to the implementation of new technologies and policy frameworks, resulting in significant progress in the long-term development of the maritime sector, as documented by several scholars [169,170]. One critical step towards this progress was the establishment of the "National Policy for the Ocean and the Coastal Spaces" (PNOEC in Spanish) by the Colombian Ocean Commission (CCO) in 2007. This policy framework plays a crucial role in the holistic integration of the oceans and their sustainable development in economic, social, and environmental aspects, shaping Colombia's maritime interests [171]. Despite this progress, the regulatory framework governing underwater cultural heritage in Colombia remains broad and has undergone several modifications over time. Furthermore, the absence of specific regulations on sunken military vessels and international cooperation poses challenges for the realization of Colombia's maritime interests.

The evolution of Colombia's regulatory framework governing underwater cultural heritage can be traced through various legal instruments, as evidenced by Lastra Mier [172], Molina Otero [173], and Cadena and Devia [166]. The earliest legal reference to shipwrecks can be found in the 1887 Civil Code (Civil Code of Colombia), which introduced the term "*especies naufragas*" ("shipwrecks", English translation). This code continues to be in force with notable amendments. Laws 14 and 36 of 1936 were the first that addressed the protection of historic assets and monuments, which was subsequently revised in Law 163, of 1959, safeguarding the extraction of cultural heritage and positioning Colombia as one of the pioneer countries in the Americas in terms of cultural heritage protection [174].

Subsequent regulations were introduced by Decree 655 of 1968, which granted the Colombia Maritime Administration (DIMAR) the right to supervise and grant permits for the search and salvage of antiques from shipwrecks within the territorial sea and the continental shelf. Decree 2349 of 1971, and later Decree 2324 of 1984, introduced further regulations on this matter, including the legal concept of "*especies naufragas*". Law 10 of 1978 incorporated Decree 264 of 1936, which regulated Law 163 of 1959, and significantly expanded the national maritime territory, by recognizing the territorial sea with full sovereignty to a distance of 12 nautical miles. Additionally, the Colombian Constitution of 1991 reinforced the commitment to heritage preservation. Article 73 of the constitution recognizes the cultural heritage of the nation and establishes that this heritage is inalienable, unseizable, and imprescriptible. The Constitution also provides a framework for the reacquisition of cultural heritage items that are in the possession of private individuals.

In 1997, the Colombian Cultural Law (Law 397 of 1997) introduced the concept of cultural heritage and extended protection to all evidence of human activities, including underwater cultural heritage. Law 1675 of 2013 represents Colombia's efforts to safeguard its underwater cultural heritage. However, this law has significant implications that negatively affect UCH, as it allows commercial salvage in alliance with the government [9]. Despite this, one notable provision of Law 1675 is the adoption of a 100-year limitation period for permanently submerged shipwrecks, in line with the criteria set forth by the 2001 UNESCO Convention [36]. This legal instrument also recognizes the value of preserving UCH in the territorial sea, exclusive economic zones, and continental shelf, extending protection beyond territorial waters and recognizing it as an intrinsic component of Colombian UCH. Furthermore, Law 1675 provides a comprehensive definition of UCH in Article 2, which encompasses all items resulting from human activities that represent culture and that are permanently submerged in internal waters, rivers, and lakes, as well as the territorial sea, the contiguous zone, the exclusive economic zone, and the continental shelf and islands, as well as other delimited areas. Law 1675 also outlines the circumstances under which items are not considered UCH, including those resulting from subsidence, shipwrecks, or evictions that have not reached 100 years from the occurrence of the event. Salvage of such items is regulated by the Commercial Code, Act 410 of 1971, and the Civil Code of 1887 Article 710, as well as other applicable national and international standards.

Despite the extensive legal framework in Colombia for the protection of cultural heritage, the law governing UCH has faced criticism. While the legislation provides guidelines for the identification, management, and preservation of UCH sites, some experts consider its protection inadequate [9]. The law allows commercial salvage under a process of public association, which permits private entities to recover historical shipwrecks and artifacts by paying 50% of the recovered elements or artifacts that are not recognized as heritage by the government [3–5]. This provision in Law 1675 has generated significant debate and controversy among various stakeholders, including the government, commercial salvage companies, and the scientific community. The University Network of Submerged Cultural Heritage (*Red Universitaria de Patrimonio Cultural Sumergido*), composed of scholars from several Colombian universities, has expressed concern about the potential illegal trade of cultural heritage [175]. Furthermore, the law has been criticized for its failure to recognize sunken military vessels in addition to its lack of provisions for international cooperation, among other deficiencies, further fueling controversy. The University Network has been instrumental in the preservation of underwater cultural heritage. They have consistently worked towards increasing public and governmental awareness regarding the urgent requirement for a dedicated state policy focused on UCH. The Network strongly advocates for the formulation and implementation of a new policy that promotes collaboration among academic sectors, social groups, and territorial actors who possess the required expertise and influence in this field [175].

However, one of the significant milestones in underwater cultural heritage in Colombia is the Constitutional Court's Judgment C264 of 2014 (*Sentencia* C-264/14). This landmark decision declared paragraphs 1 and 2 of Article 3 of Law 1675 unconstitutional due to their

contradiction with the provisions of Articles 63, 70, and 72 of the Political Constitution of Colombia. Specifically, these paragraphs failed to recognize the commercial value of raw materials, regardless of their origin, as well as serialized movable goods such as coins and ingots, which possess exchange value. Furthermore, through Resolution 0085 of 2020 (*Resolución número* 0085 de 2020), the Ministry of Culture designated the wreck site of the Spanish galleon *San Jose* as a "Cultural Heritage of National Interest" (*Bien de Interés Cultural del Ámbito Nacional*). This designation ensures the conservation and protection of this invaluable historical shipwreck. The Resolution highlights the scientific and cultural significance of the *San Jose*, emphasizing that it represents the most momentous discovery in the country's archaeology due to its remarkable state of preservation [12].

To effectively manage UCH sites and prevent the illicit commercialization of cultural heritage, it is imperative to address these concerns by revising the existing legislation comprehensively. The scientific community and civil society have echoed this sentiment and called for a reevaluation of the law to ensure that it adequately addresses these concerns [14,176].

*5.2. The United States' Domestic Laws*

The US recognized and committed its action to the importance of becoming a maritime power from an early stage, maintaining an active engagement [56,57,177], which largely explains the immense UCH heritage in oceans around the world. Due to its maritime expansion, the US is classified as a naval power with symmetrical interests in recognizing, regulating, and defending UCH, especially regarding sunken military vessels. This influence explains their initial contributions to the subject, and the enduring significance of the problem [178]. This active engagement as a naval power demands the emergence of a sophisticated and specialized maritime policy that establishes distinct advantages for the protection of UCH. The evolution of this specialized regime can be described on a variety of fronts, ranging from its sea lines of communication to the implementation, development, and evolution of an active admiralty with a broad jurisprudence in shipwreck and salvage. Furthermore, the United States' position as a naval power implies there are a great number of sunken military vessels distributed across the oceans, a continuing evolution of UCH laws that explains its prominence, and the current robust regulatory regime [35,179]. The US initiated from its Constitution in its Article 3, Section 2, to the Sunken Military Craft Act of 2004, including, the Antiquities Act of 1906; the Historic Sites Act of 1935; the Submerged Lands Act of 1953; the National Historic Preservation Act of 1966; the Federal Sovereign Immunities Act of 1976; the Archaeological Resources Act of 1979; and Abandoned Shipwreck Act of 1987 [180–183]. Nevertheless, despite these extensive statutes in place to protect UCH, an active and prominent salvage industry often contradicts these aims and regulations [184]. Given the complexity and breadth of the regulatory system, it is a matter of concern for federal courts to ensure UCH preservation [184,185]. Although states own the property of shipwrecks found along their coasts, the federal government plays a crucial role in determining shipwreck registry and ownership [185,186].

In the US, two significant acts regulate underwater cultural heritage, sunken military shipwrecks, and those with historical significance. The first is the Abandoned Shipwreck Act of 1987 (ASA) (Public Law 00–298; 43 U.S.C. §§ 2101–2106), which manages, recovers, and preserves abandoned shipwrecks and submerged cultural resources within the waters of the United States. The ASA covers and protects three categories of shipwrecks and defines historical shipwrecks based on criteria such as significant contributions to history or any other distinctive characteristic of information in prehistory or history. It is important to note that the Abandoned Shipwreck Act extends its protection to foreign shipwrecks in the territorial waters of the United States [180]. However, the act does not provide a clear definition of underwater cultural heritage [185].

The second act, the Sunken Military Craft Act (SMCA), was enacted in 2004 as part of the National Defense Authorization Act, also known as Title XIV. SMCA aims to protect and preserve the sovereign status of sunken military vessels that were operated by the

government during noncommercial military activities worldwide. It defines a sunken military craft as any sunken warship, naval auxiliary, or other vessel owned or operated by a government on military noncommercial service, any sunken military aircraft or military spacecraft owned or operated by a government when it sank, and the associated contents of such craft if title thereto has not been abandoned or transferred by the government concerned. SMCA provides immunity to all shipwrecks without exception of time and prohibits unauthorized activities that could disturb the sunken military craft, with penalties for violations intended to protect their heritage globally. However, the definition of Military Craft has been debated due to its limited scope in assessing certain types of sunken ships, such as the sunken Liberty ships that operated during World War II and sunken privateers during the Revolutionary War [187,188]. Moreover, the SMCA grants the right, title, and interest of the United States in its sunken military craft, and a permit is required for archaeological, historical, or educational purposes. It also encourages international agreements to protect sunken military craft and international reciprocity regarding sunken military vessels [35,64]. Some good examples of international cooperation spurred by the SMCA include the joint preservation of the USS *Houston* (CA-30) off the coast of Indonesia and the collaborative search for the *Bonhomme Richard*, a Revolution-era shipwreck, conducted by the US Navy and French Navy [189].

## 6. Discussion and Conclusions

### 6.1. Legal and Management Implications of the USS Kearsarge in Colombia and the United States

Colombia's maritime territory boasts the submerged remains of two US military vessels, the USS *Kearsarge* and the USS *Peacock*, which are protected under the Sunken Military Craft Act of 2004 (SMCA). Following its submergence for more than a century, the *Kearsarge* was designated as underwater cultural heritage and placed under the protection of Law 1675 in Colombia. Of particular interest is the vessel's location in Roncador Cay, within Colombian territorial waters. The small and isolated islet lies roughly 350 nautical miles south of Jamaica and 240 nautical miles east of Nicaragua, and is recognized as a marine protected area, known as Seaflower [164,190].

The USS *Kearsarge* was a 1032-ton Union Mohican-class steam sloop-of-war, best known for its role in sinking the CSS *Alabama* off the coast of France in 1864, during the American Civil War. The vessel's actions during the war earned it hero status [191].

In 1894, while serving as the flagship of the commander of the US-North Atlantic Station, the USS *Kearsarge* was involved in a military operation in Haiti and later sailed to Bluefields, Nicaragua, to safeguard American interests. However, on 2 February 1894, the *Kearsarge* ran aground on Roncador Cay at around 7 p.m. [192], and despite rescue efforts by the US government, the vessel was eventually neglected and partly looted [16]. Although the USS *Kearsarge* remained forgotten for many years, the US government eventually prioritized the protection of naval heritage, including the *Kearsarge* wreck site, towards the end of the twentieth century [193].

The analysis of the USS *Kearsarge* offers a unique opportunity to examine a sunken military vessel that is not covered by the 2001 UNESCO Convention. Additionally, its location in the territorial waters of a country that has not ratified the Convention creates a regulatory gap between the two nations, each with its own set of domestic regulations. Moreover, the historical records of the USS *Kearsarge* confirm its location in Colombian territorial waters, and the identification of the remains should be feasible due to the presence of its ordnance onboard at the time of the incident [16]. This case highlights the complexities associated with addressing sunken warships that are not covered by the 2001 UNESCO Convention, particularly the challenges that arise when dealing with countries that have different maritime interests and regulations.

### 6.2. The United States and the USS Kearsarge

The USS *Kearsarge* is widely recognized as one of the most significant shipwrecks in the Caribbean Sea, and it is considered a crucial component of the United States' underwater

cultural heritage [193]. Furthermore, the presence of human remains at Roncador Cay, particularly those of Second Class Fireman Anderson Robbins who tragically lost his life during the *Kearsarge* accident, enhances the historical significance of the wreck site, placing it on par with other notable graves. Anderson Robbins, born in 1861 in Greenville, South Carolina, enlisted in the US Navy as a Coal Heaver in 1892 and served on board the USS *Kearsarge* until his death in Roncador on 3 February 1894 [194].

The NHCC is responsible for protecting and managing the wreck site and places a high priority on international cooperation in preserving its sunken military vessels, as evidenced by existing bilateral practices [68,195]. The NHHC's Underwater Archaeology Branch (UAB) provides significant support and opportunities for scientific cooperation [63]. The long-standing positive diplomatic relations between Colombia and the United States [196] provide an opportunity for cooperation in the preservation of the USS *Kearsarge*, similar to the CSS *Alabama* protection agreement [30].

The United States Department of State has enacted various agreements, both bilateral and multilateral, to safeguard the sovereign immunity of vessels outside its jurisdiction. Among the notable examples are the RMS *Titanic*, *La Belle*, and CSS *Alabama* [35]. Interestingly, the CSS *Alabama* was sunk by the USS *Kearsarge* and was discovered and identified in 1984, seven miles from the Normandy coast near Cherbourg, by a French navy archaeological expedition [197]. Due to its location in France's territorial sea, the question of sovereign immunity and ownership arose, leading to tension between France and the United States. Nevertheless, the United States was able to establish its title based on the surrender of the *Alabama* captain to the USS *Kearsarge* and its possession of the rights to all Confederate States of America after the War [195,198]. After negotiations, both governments signed an agreement in 1989 that sought to protect the wreck site, acknowledge the title to the vessel and its artifacts, and recognize the CSS *Alabama* as shared heritage of both nations. This agreement also included provisions for scientific cooperation and archaeological research [25,30,34]. Today, the artifacts recovered from the CSS *Alabama* are carefully preserved by the Underwater Archaeological Branch of the NHHC and are available for public viewing, serving as a testament to the heritage of the American Civil War [199].

*6.3. Colombia and the USS Kearsarge*

Although the USS *Kearsarge* has been recognized as an underwater cultural heritage site in Colombia under Law 1675 of 2013, the incident is not widely known in Colombian maritime history, and the wreck site is not listed in any official database. The lack of recognition reflects a limited understanding of the consequential maritime events in Colombia, which hold significant historical and cultural value. This is particularly evident in the archipelago where Roncador Cay is located, alongside other islets and banks [200]. Preservation of the USS *Kearsarge* wreck site in Colombia presents a significant challenge, particularly in light of the feasibility of Law 1675, which excludes international cooperation and hinders maritime research and education in a country where this field is underdeveloped, but crucial [9,13,201]. It is perhaps fortunate that the site's remote location from the continent makes it difficult for looters or amateur divers to access and damage it. Additionally, the Colombian Navy controls access to most of the banks and cays of the archipelago, adding a layer of protection to the wreck site [202].

The wreck site holds great significance for Colombia due to its potential for international cooperation. Additionally, it is of paramount importance for Colombia's cultural heritage institutions, as it represents a departure from the limitations imposed by outdated legal frameworks that have impeded underwater research in recent years [9]. To ensure the long-term sustainability of its UCH, Colombia should prioritize three essential concepts: science, legislation, and technology. This should include capacity building with international agencies to enhance research development and facilitate changes in Law 1675, such as recognizing sunken military vessels and promoting international cooperation, among other important measures. By adopting this approach, the Ministry of Culture can

collaborate with the new Ministry of Science, Technology, and Innovation to develop new policies that provide comprehensive protection to the USS *Kearsarge* and other valuable underwater cultural resources. Additionally, this approach can contribute to the long-term development of education in maritime sciences and related fields, ensuring sustainable capacity building for the future protection and research of underwater cultural heritage sites in Colombia.

### 6.4. The Future of the Wreck Site of the USS Kearsarge

Colombia and the United States hold divergent views on the management of their underwater cultural heritage and sunken military vessels. Nevertheless, it is crucial to conduct a comprehensive analysis of various aspects of each case, such as military vessels involved in commercial or non-commercial activities, the causes of their loss, and, most importantly, identification. The USS *Kearsarge* was a military ship engaged in a military operation that was neglected over time due to salvage limitations, with ordnance still on board. The wreckage is situated in Colombian territorial waters, and Colombia had full sovereignty during the accident in 1894 and continues to hold sovereignty over it today [203]. Despite the protection provided by domestic laws in both countries, neither Colombia nor the United States are parties to the UNCLOS or the 2001 UNESCO Convention [12,35,37,204]. Consequently, the USS *Kearsarge* remains at risk, and cooperative efforts are necessary to protect the site in Colombian maritime territory [64,193]. The question of whether cooperation is possible in the case of the USS *Kearsarge* arises, along with the question of whether the Colombian government has a genuine interest in such collaboration. However, the USS *Kearsarge* is an appealing prospect for an international collaborative project, as it was a military vessel engaged in non-commercial activities. Thus, it is imperative to establish cooperative efforts between the flag state and the coastal state to identify the wreckage and recognize it as shared heritage of the two nations [4]. A thorough understanding of the historical and legal context is essential in reaching a consensus on the preservation and management of the site. Conducting research in situ on shipwrecks, particularly sunken military vessels in Colombian waters, is, however, challenging due to legal restrictions imposed by Law 1675, the implications of the legal case of the Spanish galleon *San Jose*, as well as the Colombian government's interests in commercial salvage [9,78]. Despite these challenges, recent bilateral agreements between the United States and other countries on sunken military vessels demonstrate a customary interest in protecting underwater cultural heritage through international cooperation. Therefore, it is in the interest of the US to preserve the USS *Kearsarge* wreck site, but any action must be taken through an understanding between the involved governments. Moreover, the United States plays a crucial role in the future of UCH worldwide by prioritizing the identification of international dynamics, including understanding the concept of state vessels and the rights of flag states to protect their underwater heritage [2]. The management of underwater cultural heritage in the case of the USS *Kearsarge* presents an asymmetric challenge for both Colombia and the United States. Colombia lacks a properly approached solution, while the United States has established and enforced a solution within its own jurisdiction and abroad. However, this issue looks difficult to resolve at the international or domestic level in the coming years. The fate of the USS *Kearsarge* wreck site remains uncertain in light of these challenges. Although Colombia has made some efforts to preserve UCH [205], there is still a tremendous effort required in reaching an agreement with flag states regarding sunken military vessels in its territorial waters to ensure proper preservation of UCH. It is crucial that Colombia maintain its pursuit of an effective strategy to safeguard underwater cultural heritage. This involves collaborating with other nations and seeking cooperation from international organizations to develop a comprehensive legal framework that will replace the current one.

**Author Contributions:** Conceptualization, methodology, validation, and resources, W.G.P., M.J., C.E.R.-G. and A.O.J.; writing—original draft preparation and editing, W.G.P., M.J., C.E.R.-G. and A.O.J.; writing—review and editing, W.G.P., M.J., C.E.R.-G. and A.O.J.; supervision, W.G.P. and M.J. All authors have read and agreed to the published version of the manuscript.

**Funding:** This research was funded by the Ministry of Education of the Republic of Korea and the National Research Foundation of Korea, grant number: NRF-2018S1A6A3A01081098.

**Institutional Review Board Statement:** Not applicable.

**Informed Consent Statement:** Not applicable.

**Data Availability Statement:** All data are available from the corresponding author on request.

**Conflicts of Interest:** The authors declare no conflict of interest.

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
