# Peer review of "Preserving Sunken Military Vessels as Underwater Cultural Heritage in Colombia: Legal Challenges and Prospects for the USS Kearsarge Wreck Site"

_heritage, doi:10.3390/heritage6060258_

Round 1
Reviewer 1 Report
The investigation is focusing on the legal status of underwater heritage in Columbia, especially American military shipwrecks. However, the article underlines that numerous military shipwrecks lying in the Colombian waters are Spanish or British. A few sentences should explain why the authors focus on the American military vessels only. Because they constitute a more recent heritage? Because the American government has a specific policy about this issue? In case, the authors are right to insist on the case of CSS Alabama, offering another good example of the problem.
The subsection title “Colombia” should be developped (l. 269).
Author Response
Dear Reviewer,
Thank you for your valuable feedback and constructive comments on our article. We greatly appreciate your positive review and insightful recommendations.
In response to your comments, we have explained and emphasized the significance of the USS Kearsarge case and made appropriate revisions to highlight its unique analysis in the discussion section.
Furthermore, we have carefully considered your recommendation to develop the "Colombian" perspective in our research. As a result, we have expanded upon this aspect in our discussion section, providing a more comprehensive understanding of the implications and relevance of our findings.
Once again, we sincerely appreciate your time and effort in reviewing our article. Your insightful comments have greatly contributed to its improvement.
Thank you for your consideration.
Sincerely,
William Gomez Pretel
Reviewer 2 Report
It is an interesting and necessary study in the current context. It consciously addresses a controversial issue and does so with rigor and objectivity, which is to be welcomed.
The bibliography is complete, varied, and up-to-date. The structure is clear, and the conclusions are interesting. However, perhaps the main contribution is the orderly exposition of the framework of the related challenges.
It is also to be noted the approach from the different states' perspectives.
Author Response
Dear Reviewer,
Thank you for your positive feedback on our study.
We appreciate your recognition of its relevance, rigorous approach, and a comprehensive bibliography. The orderly exposition of the related challenges and consideration of different states' perspectives were important aspects of our research. Your feedback has been invaluable to us.
Thank you for your time and evaluation.
Best regards,
William Gomez Pretel
Reviewer 3 Report
The paper constitutes and important contribution as it suggests the best avenue for research is international cooperation and as it urges Colombia to adopt a legislation on flagships as UCH. However major clarifications and corrections are required to improve the paper for a Q1 publication quality.
- Law 1675 might be milestone as a law to perpetuate treasure hunting in territorial waters, but represents a serious drawback in relation to previous legislation regarding cultural heritage in general, which recognized the concept of archaeological context entailing the protection of the whole sites with the full of their components. In a journal called heritage, this should be made really clear.
- Criticism by the Colombian and the international community of scholars should be definitively further developed, explaining the terrible implications of this law for UCH and for cultural heritage in general. Maybe a Sentencia C264-14, which is not even mentioned, and declares Article 2. inexequible fighting against (representatividad y unicidad) and the declaration of the San José as Bien de Interés Cultural might be milestones in the protection of UCH. Sentencia C264-14 needs to be addressed and reviewed.
- Law 1675 is far away from being in accordance to the UNESCO Convention of 2001. This must be rephrased to explain it goes exactly in the opposite direction.
- It is ill-informed to assert Colombia holds no official inventory of UCH. Several efforts have been made in this direction, some of them with cooperation of Foundations Eriagaie and Terra Firme and the ICANH. The big problem is that they cannot be accessed by the general public. Not even by academics aiming to find a site to research.
- It is very ambitious to say the list of shipwrecks is a comprehensive list. It is just a list, which contributes but requires much more research to be comprehensive. The lists should be complemented by cartographical work and historical or archaeological data.Cartography must be included, at least a map.
- The issue of what is cultural cargo and what is precious cargo, should be readdressed to explain they are both cultural materials that are part of the archaeological context which encompasses all the contents of the wreck and the surrounding environment.
- I suggest to better review the work of Garcia and Del Cairo, and specifically the large amount of work of Del Cairo and his reflections on Maritime Cultural Heritage and its management through interinstitutional cooperation and participation of stakeholders. The work of McKinnon in Saipan with stakeholders is very important and should be considered. Beyond the State’s and Governamental interests the emphasis should be placed on the stakeholder of cultural Heritage and the general public. I suggest to check as well Borrero’s paper on war vessels and battlefield archaeology, to provide a general overview of the archaeological work related to this type of sites.
- Even more emphasis should be placed on the importance of international cooperation and the community participation (stakeholders).
- The work conducted on D-Day battle field and the work conducted by SAS-INAH in Mexico with US submarine H1 is worthed of mention, as it involves international cooperation with retired officers of the US Navy.
- La Belle is a famous French Shipwreck of la Salle Expedition. If there is a la Belle of the US Navy this should be clarified.
- Laws should be cited as primary sources, not in the bibliography.
The English is fine.
Author Response
Dear reviewer, we truly appreciate your valuable comments, which have undoubtedly enriched our article and helped clarify certain aspects. We have carefully reviewed your suggestions and made the necessary corrections throughout the paper to ensure its accuracy and clarity. Your input has been invaluable in improving the quality of our work.
1. Law 1675 might be milestone as a law to perpetuate treasure hunting in territorial waters, but represents a serious drawback in relation to previous legislation regarding cultural heritage in general, which recognized the concept of archaeological context entailing the protection of the whole sites with the full of their components. In a journal called heritage, this should be made really clear.
Response:
Thank you for your valuable review. We acknowledge and fully agree with the reviewer's assessment of the critical and negative aspects of Law 1675. We also appreciate their feedback on the parts referred to as "milestones" in the paper. As a result, we have made the necessary amendments and corrections to improve the overall quality of the paper. Your input has been instrumental in refining our work.
2. Criticism by the Colombian and the international community of scholars should be definitively further developed, explaining the terrible implications of this law for UCH and for cultural heritage in general. Maybe a Sentencia C264-14, which is not even mentioned, and declares Article 2. inexequible fighting against (representatividad y unicidad) and the declaration of the San José as Bien de Interés Cultural might be milestones in the protection of UCH. Sentencia C264-14 needs to be addressed and reviewed.
Response:
Thank you for your comments. We have expanded the work of the University Networking as initially suggested. Additionally, we have included and explained the significance of Sentence C-264-14 and the galleon San Jose declaration as a cultural heritage site of interest. These additions and elaborations have further enriched the content of our paper. We are grateful for your input and believe that these enhancements have strengthened the overall academic value of our research.
3. Law 1675 is far away from being in accordance to the UNESCO Convention of 2001. This must be rephrased to explain it goes exactly in the opposite direction.
Response :
Thank you for this comment. We have taken the necessary steps to clarify the points raised in our research. The information has been thoroughly reviewed and revised to ensure accuracy and clarity.
4. It is ill-informed to assert Colombia holds no official inventory of UCH. Several efforts have been made in this direction, some of them with cooperation of Foundations Eriagaie and Terra Firme and the ICANH. The big problem is that they cannot be accessed by the general public. Not even by academics aiming to find a site to research.
Response:
We appreciate your comment and would like to clarify that during the course of our research, we contacted several agencies, including ICANH and DIMAR, regarding the establishment of a database. While only DIMAR responded, informing us about the construction of a database, we acknowledge that this issue needed further clarification. We have made the necessary correction based on your feedback. Thank you for bringing this to our attention and helping us improve the clarity and accuracy of our work.
5. It is very ambitious to say the list of shipwrecks is a comprehensive list. It is just a list, which contributes but requires much more research to be comprehensive. The lists should be complemented by cartographical work and historical or archaeological data.Cartography must be included, at least a map.
Response:
Thank you for your comments. We have made the necessary corrections by changing to "list" in our article. However, we acknowledge that we did not include maps depicting the locations of the shipwrecks during the construction of the article. This decision was primarily due to the ambiguity surrounding the precise locations of several shipwrecks mentioned in the lists.
6. The issue of what is cultural cargo and what is precious cargo, should be readdressed to explain they are both cultural materials that are part of the archaeological context which encompasses all the contents of the wreck and the surrounding environment.
Response:
Thank you for bringing up this comment. We have provided a more thorough and elaborate explanation of the topic within the article. Additionally, we have identified certain parts where the inclusion of "precious cargo" occurred and have deleted them to enhance the clarity and focus of the content.
7. I suggest to better review the work of Garcia and Del Cairo, and specifically the large amount of work of Del Cairo and his reflections on Maritime Cultural Heritage and its management through interinstitutional cooperation and participation of stakeholders. The work of McKinnon in Saipan with stakeholders is very important and should be considered. Beyond the State’s and Governamental interests the emphasis should be placed on the stakeholder of cultural Heritage and the general public. I suggest to check as well Borrero’s paper on war vessels and battlefield archaeology, to provide a general overview of the archaeological work related to this type of sites.
Response:
Thank you for your comment. We acknowledge the valuable contribution of Del Cairo in the field, and we have duly recognized and cited their work throughout the paper. We appreciate your suggestion regarding the inclusion of additional articles related to the topic. However, it's worth mentioning that our research encompasses a comprehensive range of sources, including over 200 citations from various sources in both Spanish and English languages. These citations have been carefully selected to support and substantiate our statements and arguments.
8. Even more emphasis should be placed on the importance of international cooperation and the community participation (stakeholders).
Response:
Thank you for your comment. We have indeed emphasized and elaborated on the significance of international cooperation and stakeholder engagement in our paper to underscore their crucial role in the preservation of underwater cultural heritage.
9. The work conducted on D-Day battle field and the work conducted by SAS-INAH in Mexico with US submarine H1 is worthed of mention, as it involves international cooperation with retired officers of the US Navy.
Response:
Thank you for this comment. However, throughout the article, we extensively acknowledge the work conducted by the Naval History and Heritage Command and other relevant agencies in the United States. We have dedicated a significant portion of the paper to highlighting and providing examples of their valuable contributions and efforts in the field of underwater cultural heritage preservation.
10. La Belle is a famous French Shipwreck of la Salle Expedition. If there is a la Belle of the US Navy this should be clarified.
Response:
Thank you for your comment. We acknowledge the distinction between "La Belle" and USS "Belle", and we have consistently used the appropriate acronyms, such as "USS," when referring to ships from the US Navy in our paper. Furthermore, the provided citation serves to clarify this distinction.
11. Laws should be cited as primary sources, not in the bibliography.
Response: We appreciate your comment. We have made sure to cite Colombian domestic laws and provide a bibliography to ensure that readers have easy access to the relevant legal sources. This approach aims to prevent any confusion and allows readers to refer to the laws directly for further clarification.
Reviewer 4 Report
Although the article analyses an interesting aspect of the UCH, it is very interesting and could bring an important contribution to the legal side of the management of sunken state vessels, not just the military. However, I do not agree with how the authors approach the legal status of either the State or Military wrecks. It must be completely rewritten in my opinion.
State vessels and even more military vessels are well-defined in UNCLOS. However, UNCLOS does not cover the vessels after their loss. It doesn’t matter what activity they were conducting. They could only be conducting Scientific research, for example.
To cover that gap, both the UK and the USA created their own State/Military UCH protection laws (Protection of Military Remains Act 1986 (UK) and the Military Craft Act of 2004 (SMCA) (TSA).
However, being national laws, other countries are not obliged to follow them. Unfortunately, we have seen examples of this both in the Jutland wrecks and in the Far East, where both WWI and WWII have been plundered.
That was why the UNESCO 2001 convention was created.
And, when countries have not signed the Convention, they can and should establish bilateral agreements between both or several parties, as the authors show in the overall article.
Author Response
Dear Reviewer,
Thank you for taking the time to review our article. We appreciate your valuable feedback and insightful comments.
We have carefully considered your suggestions and have made significant improvements to the article. Specifically, we have addressed the issue regarding the legal status of sunken vessels, as you highlighted, and have emphasized the lack of coverage by the UNESCO Convention after their loss. Additionally, we have incorporated your comments into the section discussing UNCLUOS, resulting in a more comprehensive analysis.
Your comments have played a crucial role in enhancing the quality and clarity of our article, and we greatly value your thoughtful evaluation.
Best regards,
William Gomez Pretel
Round 2
Reviewer 3 Report
A map should be added, clarifying that the locations of the wrecks is approximate and not precise.
Englis is fine.
Author Response
Dear Reviewer,
We would like to express our sincere gratitude for taking the time to read our manuscript. Your insightful comments have been immensely helpful in improving the article.
In response to your suggestions, we have made significant revisions. Specifically, we have followed your advice to cite laws from primary sources rather than in the bibliography. Furthermore, we have incorporated two articles by Del Cairo and another relevant source that discuss the cooperative efforts between stakeholders and state agencies in Colombia. These examples serve as valuable illustrations of how such collaboration can effectively protect and promote future projects related to military sunken vessels in Colombia.
Comment 1. A map should be added, clarifying that the locations of the wrecks is approximate and not precise.
Response:
Our main objective was to address the legal gaps in the Colombian regulatory framework pertaining to military shipwrecks, using the USS Kearsarge as a case study. We included a "list" of military sunken vessels to provide an overview of the naval heritage derived from such wrecks in Colombia, considering the absence of an official and easily accessible comprehensive list.
Regarding the suggestion of including maps with approximate locations, we understand their potential value. Despite the aforementioned challenges posed by the ambiguous nature of the available information and reliance on secondary references, our approach has been to address this issue by including specific geographic location names along with their respective countries in the list of military sunken vessels. In order to provide clarity regarding the geographical context, we have now mentioned "Colombia". As for the USS Kearsarge, we have taken the opportunity to clarify that it is located in Colombian territorial waters near Roncador Cay. This clarification is based on a previous article published in the journal "Heritage" that utilized maps and GIS technologies to estimate the wreck site.
We fully agree that creating a comprehensive and accurate map would require a separate project with access to primary sources such as logbooks, and further research. Nevertheless, we believe our current work can serve as a foundation for future endeavors in that direction.
We appreciate your understanding and valuable feedback.
Reviewer 4 Report
I think the authors now have nailed the issue, that is really complicated. Well done
Author Response
Dear Reviewer,
Thank you for your positive feedback and recognition of our efforts in effectively addressing the complex issue at hand. We greatly appreciate your kind words and encouragement.